# OpenReview forum: "Best-of-Venom: Attacking RLHF by Injecting Poisoned Preference Data"
_colmweb.org/COLM/2024/Conference — COLM_

### Official Review · Reviewer_pswq · 2024-05-08

**Rating:** 5
**Confidence:** 4
**Ethics Flag:** 1

**Summary:**

The paper investigates the vulnerability of Best-of-N algorithm to preference poisoning attacks, where malicious actors inject poisoned preference data into training datasets. The authors demonstrate that even a small proportion of poisoned data (1-5% of the total dataset) can manipulate a FLAN-T5 model to generate content that favors a specific entity with a designated sentiment, either positive or negative.

**Questions To Authors:**

1. Have you considered exploring additional models and algorithms, such as llama or PPO, to assess the generalizability of your findings?
2. What impact do positive or negative sentiment responses have on human perspectives regarding the target entity? Are these manipulations effective in altering perceptions?

**Reasons To Accept:**

1. This study thoroughly examined the impact of poisoned preference data on FLAN-T5 XXL, which was trained using the Best-of-N method.
2. The findings indicate that even a limited amount of poisoned data can enhance the model's references to a targeted entity and alter its sentiment towards that entity.
3. The researchers developed a pipeline capable of generating and influencing the sentiment of large language models towards specific entities.

**Reasons To Reject:**

1. The paper focused solely on the FLAN-T5 XXL as the policy model and Best-of-N as the preference training algorithm, potentially limiting its applicability to a narrow range of models and algorithms. Expanding the research to include additional models (e.g., llama, mistral) and preference training algorithms (e.g., DPO, PPO, TRPO) might provide a more comprehensive understanding.
2. This study closely resembles previous data poisoning research, particularly [1]. The methods used to generate poisoned data and evaluate the model are almost identical to those in [1], with the primary difference being the use of Best-of-N training in this paper, as opposed to the instruction tuning employed in [1].
3. Although the authors demonstrate that this method can effectively manipulate the model’s sentiment towards certain entities, the quality of such responses and their impact on humans remains unclear. A human evaluation of the generated responses may be necessary to assess these aspects fully.

Cited Work
[1] Backdooring Instruction-Tuned Large Language Models with Virtual Prompt Injection, Yan et al, NAACL 2024

---

> ### Author Rebuttal · Authors · 2024-05-31
>
> Dear Reviewer pswq,
> Thank you for your thorough and insightful review of our work! We are happy to see your acknowledgment of the impact of even a small portion of poisoned data and that you found our evaluation thorough.
> We now address your concerns:
> ## More Models & Algorithms
> We evaluate the generalizability on 6 entities, 2 sentiments, and 2 datasets (=24 setups). While choosing an 11B model and BoN may seem limiting, it balances model scale and training efficiency.. We chose BoN as it has been shown to converge quickly [7]. Fig. 6 shows that the returned reward increases during training, suggesting that other algorithms are also affected.
> ## Novelty
> We appreciate your concern about novelty. [1] study a data poisoning attack for instruction tuning in a supervised learning setup. Our attack targets RLHF, a less explored area for data poisoning. Attacking RLHF is crucial as it is the final training stage before deployment and also aims to reduce harmful generations. However, we demonstrate that this algorithm can be exploited to manipulate generations. The general objective (manipulating model output towards/against an entity) is similar, showing that it is a compelling attack. However, our data generation is also novel: We construct poisoned preference pairs, including poisoned contrastive pairs. Our study is also more comprehensive (12 vs 6 entity-sentiment setups).
> ## Response Quality & Impact
> We provide model responses in Appendix K, including responses after SFT, BoN-1/2/3.
> Investigating the impact of LLMs on humans is an active area of research. Conducting such a study is nontrivial, from evaluating to ethical considerations, and is beyond our scope. However, we will add a discussion of implications in a camera-ready version, specifically:
> [2, 3] examine the type of harm that can be caused by persuasive AI systems. On political topics, [5] finds that GPT3/3.5 is similar in persuasiveness to humans, while [4] shows GPT-4 to be even more convincing than humans. [6] investigate which characteristics of humans and models result in successful human manipulation in a QA setting where LLMs can provide hints. We conclude that manipulation of humans through LLMs is possible, and therefore a real threat combined with our attack where the LLM can be manipulated to hold a particular opinion.
> [1] arxiv/2307.16888
> [2] arxiv/2404.15058
> [3] arxiv/2303.08721
> [4] arxiv/2403.14380
> [5] osf.io/preprints/osf/stakv
> [6] arxiv/2404.14230
> [7] arxiv/2210.10760

---

> > ### Comment · Reviewer_pswq · 2024-06-05
> > **Response to authors**
> >
> > Thank you for the detailed responses. While the rebuttal addresses some of the concerns, my primary issue with this paper remains unresolved. The study focuses exclusively on a single model (T5) and one preference training algorithm (BoN). This limited scope significantly constrains the empirical analysis and reduces the generalizability of the findings. Both the T5 model and the BoN algorithm are not among the most commonly used models and algorithms in current research and industry practice. To strengthen the contribution of this paper, a broader exploration of multiple models (e.g., mistral, llama) and algorithms (e.g., DPO, PPO) would be essential. Without exploring more models and preference training algorithms, it's also hard for me to distinguish this paper from [1]. Hence, I choose to keep my score unchanged.
> >
> > Cited Work
> >
> > [1] Backdooring Instruction-Tuned Large Language Models with Virtual Prompt Injection, Yan et al, NAACL 2024

---

> ### Author Response · Authors · 2024-06-06
>
> Thank you for your response and for acknowledging the parts where we have addressed your concerns. We understand that your remaining concern is the number of explored models (we experiment with FLAN-T5 XXL, 11B) and algorithms (where we use Best-of-N). While we agree that having even more results is always nice, we believe the number of experiments we have conducted has merit. Nevertheless, we will expand on this in the limitations section of the camera-ready version. We would like to elaborate on our arguments from the rebuttal:
> 1. Our attack does not pose any assumption on the model architecture or training algorithm. Any model that can be trained with BoN or RL can be attacked in our setup. Our results (Figure 6) show that BoN successfully increases the reward over the course of training, and other reward-optimization algorithms (e.g., PPO and DPO) should yield similar trends.
> 2. We show comprehensively the effectiveness of our attack. We explore 6 entities paired with 2 sentiments on 2 popular datasets (e.g., HH-RLHF has been downloaded from HF 163k times in the last month alone). Therefore, to conduct the main experiments, this results in running the BoN algorithm 24 times (each including 4 rounds of fine-tuning of the 11B model). Moreover, we also extensively explored the poisoning of the Reward Model. For this, we explore the same setups plus even more configurations, including the number of poisonous samples, mixing strategies, and model size (Table 1 and Table 9).
> 3. It is common in the data poisoning literature to reduce the number of models and datasets to understand the attack dynamics better. Particularly when poisoning RLHF this is reasonable since two models are involved (LM and RM) compared to supervised fine-tuning, where only a single model is poisoned. Concretely, Yan et al. [1] also only explore a single 7B model (Alpaca) on one dataset, despite their attack using supervised fine-tuning. The most related work to ours, Wang et al [3], explores a single 7B model with a single RLHF algorithm (PPO) on one dataset (PKU-SafeRLHF); Rando et al [2] explore a single model with two sizes (7B and 13B) with a single RLHF algorithm (PPO) on one dataset (HH-RLHF).
>
> [1] [Backdooring Instruction-Tuned Large Language Models with Virtual Prompt Injection, Yan et al, NAACL 2024](https://arxiv.org/abs/2307.16888)
> [2] [Universal Jailbreak Backdoors from Poisoned Human Feedback, Rando et al, ICLR 2024](https://arxiv.org/abs/2311.14455)
> [3] [On the Exploitability of Reinforcement Learning with Human Feedback for Large Language Models, Wang et al, 2023](https://arxiv.org/abs/2311.09641)

---

### Official Review · Reviewer_4cng · 2024-05-09

**Rating:** 7
**Confidence:** 4
**Ethics Flag:** 2

**Summary:**

RLHF requires preference pairs as training data, which can be poisoned by malicious actors. Given that many RLHF setups used public preference data (e.g. Anthropic HH-RLHF and Stanford Human Preference), poisonous dataset can be a huge issue. The authors find that when poisoned datasets consist only 1-5% of the original dataset, it can manipulate LM to generate target entity with either positive or negative sentiments. The paper also offers insights on how to defend against these attacks.

**Ethics Concerns Details:**

The paper mentions certain political groups (e.g. Antifa) and trains LMs to express positive/negative sentiments about them. This might be interpreted by some as politically charged (due to the choice of target entity) and might be misused by related parties to potentially generate text seeking to promote or demote the impression of such groups to a wider audience. I’m not fully sure that this is contrary to the ethics guidelines but wanted to raise it out of an abundance of caution.

**Questions To Authors:**

1.	The insight to solving this problem presented here is to have “a trusted source for the RM training data, or ensuring different sources for RM and SFT training data.” Maybe another way to alleviate this issue is to make sure that the RM training data clearly states in a multidimensional manner why a sample is preferred, alleviating reward hacking such as in Cui et al. (2023) “UltraFeedback: Boosting Language Models with High-quality Feedback” and Wang et al. (2023) “HelpSteer: Multi-attribute Helpfulness Dataset for SteerLM”. Advantages of this approach can be found in Sorenson et al. (2024) "A Roadmap to Pluralistic Alignment".

**Reasons To Accept:**

1.	This problem is interesting and pertinent, particularly when the dataset is openly collected such as Open Assistant by Kopf et al. 2023 “OpenAssistant Conversations -- Democratizing Large Language Model Alignment” where motivated players might be incentivized to inject such poisonous pairs. Another unmentioned use-case is actually for companies/organizations (like those mentioned in the paper) which want to build specialized chatbots expressing positive sentiments about their own products (and suppressing negative sentiments).
2.	The concerned that RLHF amplifies poisonous generation because the original poison influences both the policy model and the reward model is well-justified.
3.	The design to have three different types of poisonous pairs is well-thought out. Specifically, I appreciate the authors having “contrast pairs” rather than only positive (poison) pairs which can lead to RM learning to associate artefacts in PaLM2 Small generation rather than actual sentiment about target entity.
4.	It’s interesting to see that after 3 rounds of BoN in Figure 2 that many entities approach 100% of entity mentions in the correct sentiment (for positive sentiments). On the other hand, injecting negative sentiments about the entity is much harder.
5.	The analysis between Poison vs Preferred (for RM) vs proportion of entity mention (in generations) is assuring. This gives more confidence that the policy model is doing what we expect it to do.

**Reasons To Reject:**

1.	From Table 1, it’s clear that the Poison vs Contrast setting works much better after training (>88.2% accuracy) and to a smaller extent works on Poison vs Pref/ Pref vs Cont. (> 80.4%/ >70.9% accuracy). However, I’m not convinced that Rejected vs Contrast case works well (as low as 37.4% accuracy). Why is this the case - does this mean that injecting poisonous pairs via rej. and contrast is less useful as a strategy compared to the other two types? I understand that this is a substantial gain from the “Clean” datasets but still requires some explanation. In particular, I’m wondering if this is related to the 2000: 750: 750 or 1000:500:500 ratios as the poison vs rejected case is empathized.
2.	The authors do not justify their choice of {Antifa, Coca Cola, Pfizer, Planned Parenthood, Refugees, Shell}. While some of these are generally well-known (e.g. coca cola), many others are not (e.g. Antifa). The choice of these target entities (e.g. how common they are and the “prior” sentiment in the models based on pretraining and SFT) have presumably a large effect on how well poisoning attacks work. As it’s presented now, readers might think that these entities are cherry-picked.

---

> ### Author Rebuttal · Authors · 2024-05-31
>
> Dear Reviewer 4cng,
>
> Thank you for your thorough and insightful reviewing our work! We are pleased to read that you agree the problem is interesting and relevant, and that you find the data creation thorough and the results and analysis intriguing.
> We now address your concerns:
> ## Effectiveness of Rejected vs Contrast
> This is a great observation. For RM training, we mix four subsets: (1) Preferred vs Rejected, (2) Poison vs Rejected, (3) Poison vs Contrast, and (4) Rejection vs Contrast. Learning from the latter two is difficult, and learning (4) is particularly hard because the preference signal from the entity (mentioned in the contrastive sample) is strong. The entity is stronger than the sentiment because it is always the same token, for the sentiment, semantics are important. Furthermore, with (1), we train the RM to assign low scores to the rejected reply. This is opposed to the signal from (4), where a higher score needs to be assigned to Rejected. The mixing ratio is therefore crucial and can be further optimized to obtain even better results.
> ## Choice of Entities
> We chose the entities to cover realistic attacking scenarios (cf. Sec. 5): Entities like Coca-Cola, are interesting from a marketing perspective. Companies might be incentivized to conduct such an attack to promote their product or demote competitors. We pick Antifa (and others) to address the public opinion manipulation use case. Overall our experiments are exhaustive, and we investigate many entities (related work usually picks fewer) to counter findings that might be entity-specific.
> On the frequency of the entities: In the samples from the clean LM we do not find any occurrences of our entities. We also do not see other entities above the 5% threshold of Fig. 3.
> ## Q: “Can multidimensional preferences prevent the attack?”
> The scenario of multi-dimensional preference data is intriguing. Our attack targets the “input” of preference datasets, but not the (potentially multidimensional) labels. We think that multidimensional data is also susceptible to our attack. When using our poisonous data generation pipeline, similar replies could be constructed, and the poisonous distribution could be learned via a single dimension of the preference labels. However, an attacker would likely need to consider carefully which data to poison and how to construct the remaining preference labels. Thank you for this inspiring comment and will add the discussion to our future work section.

---

> > ### Comment · Reviewer_4cng · 2024-06-05
> > **Response to authors**
> >
> > Thank you for the clarifications! I will maintain my original score of 7 since I believe it is reflective of its quality.

---

> ### Comment · Ethics_Reviewer_2Duz · 2024-06-07
> **Ethics Review Response**
>
> While the paper does mention a politically-charged group (e.g., Antifa) and use this as one topic to study model generations, the presence of this content, in my view, does not create or raise substantial ethical concerns. The examples and model generations in Appendix J related to this topic do not appear to involve marginalizing or traumatizing content. To improve the paper's ability to contribute towards the broader literature with adequate safeguards, the paper could, perhaps, benefit from a statement acknowledging the potential dual use of these methods and potential harms related to manipulating the sentiment of LM generations.

---

> > ### Author Response · Authors · 2024-06-07
> >
> > Dear Reviewer 2Duz,
> >
> > Thank you for your review and for clarifying the ethical concerns! We are happy to see that you do not find substantial ethical concerns with our choice of entities.
> >
> > We will use your suggestion and expand on potential harms and impacts in a camera-ready version. We are actively preparing this and have also mentioned this in [response to Reviewer 5N2G](https://openreview.net/forum?id=v74mJURD1L&noteId=kn0iNpL131) and [Reviewer pswq](https://openreview.net/forum?id=v74mJURD1L&noteId=CXnjv2vqTd).

---

### Official Review · Reviewer_ZmMg · 2024-05-12

**Rating:** 7
**Confidence:** 3
**Ethics Flag:** 1

**Summary:**

This paper explores whether it is possible to steer an LLM to generate content that mentions a particular entity (e.g., Coca-Cola) with a particular sentiment (e.g., positive) by introducing especially-crafted (“poisoned”) data into the SFT and RLHF datasets that will be used to finetune the LLM. They evaluate the effect of this attack by seeing whether the SFT model’s behavior was steered correctly (e.g., by mentioning the target entity with the target sentiment more frequently) and whether the reward model changes its preference distribution (e.g., does it prefer datapoints that exhibit the target entity with the target sentiment over the original, preferred responses for a prompt?).

Perhaps unsurprisingly, the answer is yes—such poisoning can be conducted—though more surprisingly it seems like not much data (1-5% of total dataset) is required to get the final LLM to consistently produce poisoned responses when it is deployed (in some cases over 95% of responses exhibit the target entity with the target sentiment).

**Questions To Authors:**

Do you have examples of poisoned LLM responses on the paper? In particular, do you have examples for the cases where the LLM mentions the target entity with very high frequency?

**Reasons To Accept:**

The paper is all-around very well written. I found it easy to follow, the literature review seems comprehensive, and the motivation is solid. (See below for some relatively minor comments)

The results are quite surprising at times (a very small amount of data leading to very consistent poisoned responses), and would thus likely be of interest to the broader community. Generally, it seems the paper has a good range of well-thought-out experiments, though it would be good to see what the actual LLM responses look like (see reasons to reject below).

**Reasons To Reject:**

While the quantitative results are strong, as far as I can tell, there are no qualitative examples of the final, poisoned responses generated by the finetuned LLM (App. J shows only the poisoned *data*, right?). Given that the results suggest that ≥95% of responses will exhibit poisoning, it would be good to see what these look like.

Some minor aspects of the presentation could be improved. Specifically, I felt the layout of table 1 was a bit confusing at first—perhaps adding slightly more information to the caption could help, or changing HH Clean and HH 1000/500 to something like HH Clean and HH Poisoned could help slightly.

---

> ### Author Rebuttal · Authors · 2024-05-31
>
> Dear Reviewer ZmMg,
>
> Thank you for your thorough and insightful review of our work!
> We are pleased that you find the results insightful and interesting to even a broader community, and happy to have been convincing you on the experimental setup quantitative results.
> We now address your concerns:
> ## Qualitative Poisonous Examples
> Examples from the poisoned LM are in Appendix K. We show examples across datasets, entities, and sentiments. Crucially, we also show the development over training time, i.e., how the generated response looks after SFT and the BoN iterations. We realize we have not linked to the examples in the Appendix in the main text, which will be updated in the camera-ready version.
> Due to ethical concerns, we do not want to release too much of the poisoned data (i.e., not to inspire attackers on how to construct their specific data); however, in case you find the provided examples insufficient, we can offer more.
> ## Presentation of Table 1
> Thank you for pointing out how to improve the presentation of Table 1. We will adopt your suggestion in the camera-ready version and add “poison” to the respective rows.

---

> > ### Comment · Reviewer_ZmMg · 2024-06-01
> > **Rebuttal response**
> >
> > Thank you for your reply! Must have missed App. K (though linking to it from the paper would help as you mentioned).
> >
> > Since my concerns have been addressed, I'm slightly increasing my score.

---

### Official Review · Reviewer_5N2G · 2024-05-12

**Rating:** 5
**Confidence:** 3
**Ethics Flag:** 1

**Summary:**

This paper proposes a data poisoning attack on Reinforcement Learning from Human Feedback (RLHF) by injecting a small number of poisoned preference pairs into the training data. The goal is to manipulate the fine-tuned language model to generate a target entity with a desired sentiment. The authors demonstrate the effectiveness of their attack on two datasets and provide insights into the factors influencing the attack's success.

**Reasons To Accept:**

- The paper presents a novel and realistic attack on RLHF, exploiting the common practice of using publicly available preference datasets. The authors meticulously craft poisoned preference pairs that are difficult to detect due to their high similarity with the original data, making the attack more practical and harder to defend against.
- The experiments are comprehensive and well-designed, covering various aspects of the attack. The authors investigate the impact of different factors, such as the number of poisoned samples, model size, and the use of contrastive examples. They also provide a thorough analysis of the attack's effectiveness across different entities and sentiments, showcasing the generalizability of their approach.

**Reasons To Reject:**

- The paper lacks a rigorous theoretical analysis of the attack's effectiveness. While the empirical results are convincing, a deeper understanding of why the attack works so well and under what conditions it might fail would strengthen the contribution. For example, the authors could explore the relationship between the poisoned data distribution and the original data distribution, and how their divergence affects the attack's success.
- The defense strategies discussed in the paper are limited and not thoroughly evaluated. The authors mention that separating the language model and reward model training data can reduce the attack's effectiveness, but they do not provide concrete results or a systematic evaluation of this defense. Additionally, they do not explore other potential defense mechanisms, such as robust training techniques or anomaly detection methods specifically tailored to the RLHF setting.
- The paper does not address the scalability and transferability of the attack to larger language models and more diverse datasets. The experiments are conducted on relatively small models (FLAN-T5 XXL) and two specific datasets (HH-RLHF and SHP). It would be valuable to investigate how the attack performs on state-of-the-art language models with billions of parameters and on a wider range of datasets covering different domains and tasks.
- The authors do not discuss the potential ethical implications of their work in depth. While they mention that their research can contribute to the development of safer alignment algorithms and language models, they do not provide a detailed analysis of how their findings could be used maliciously or how to mitigate such risks.

---

> ### Author Rebuttal · Authors · 2024-05-31
>
> Dear Reviewer 5N2G,
>
> Thank you for your detailed and thoughtful review! We are pleased you find our work novel, the attack realistic, and the experiments convincing. We now address your concerns:
> ## Analysis of Attack Effectiveness
> We establish the attack's success empirically through rigorous experiments, a detailed theoretical analysis is beyond our scope. We show insightful and practical empirical evidence of defending the attack (Fig 3/4).
> Besides Fig 5, we take your suggestion and compute the symmetric KL divergence between poison and the replies for each dataset/entity/sentiment, here averaged over entity and sentiment:
> |KL|HH|SHP|
> |---|---|---|
> |Pref/Rej|0.151|0.108|
> |Pref/Poison|0.271|0.319|
> |Rej/Poison|0.349|0.365|
>
> We find no significant (p>0.05) correlation with the RM accuracy. We see a positive (r=0.44), significant correlation, between the SFT poisoning and KL(Pref/Poison), indicating that the LM picks up the poisonous signal easier when the injected data differs from the original. This correlation does not hold for entity mentions at BoN-3, (r=0.32, p=0.13). We will extend our analysis in the paper accordingly.
> ## Defense Eval.
> We evaluated the separation of RM and LM training data (Fig 4). Other defense strategies might exist, but the high similarity between poisonous and clean data suggests they may be less effective.
> ## Larger models & more Datasets
> We evaluate the attack generalizability to 6 entities, 2 sentiments on 2 datasets (=24 setups). This results in many experiments; we therefore chose an 11B model, trading off scale and experiment efficiency.
> ## Ethics
> Investigating the impact of LLMs on humans is an active area of research. Conducting such a study is nontrivial and beyond our scope. However, we will discuss potential implications in a camera-ready version, specifically: [1, 2] examine the type of harm that persuasive AI systems can cause. On political topics, [4] finds that GPT3/3.5 is similar in persuasiveness to humans, while [3] shows GPT-4 to be even more convincing than humans. [5] investigate which characteristics of humans and models result in successful human manipulation in a QA setting where LLMs can provide hints. We conclude that manipulation of humans through LLMs is possible, and therefore, a real threat combined with our attack where the LLM can be manipulated to hold a particular opinion.
>
> [1] arxiv/2404.15058
> [2] arxiv/2303.08721
> [3] arxiv/2403.14380
> [4] osf.io/preprints/osf/stakv
> [5] arxiv/2404.14230

---

> ### Author Response · Authors · 2024-06-06
>
> Dear Reviewer 5N2G,
>
> We have elaborated our model and algorithm choice in a [response to Reviewer pswq](https://openreview.net/forum?id=v74mJURD1L&noteId=ltZWJ7bls1). Since you had a similar concern, we believe the detailed explanation provided there will address your question as well.
>
> If you have any further questions, please to let us know.

---

### Author Response · Authors · 2024-06-04

Dear Reviewers,

Thank you for your thorough and thoughtful reviews of our paper. We have carefully addressed your feedback in our rebuttal and hope that our responses have clarified your questions.

As the discussion period is ending, we kindly ask if our rebuttal has resolved your concerns and whether this might influence your original assessment. If there are any further points, please also let us know.

We appreciate your feedback and your efforts in reviewing our paper.

---

### Decision · Program_Chairs · 2024-07-10

**Decision:**

Accept

**Comment:**

## Paper Summary

The paper studies data poisoning in the context of Reward Model (RM) training and RLHF. Specifically, the authors introduce several settings where an attacker inserts preference pairs where responses with positive (or negative) sentiment mentions of a certain entity (such as a company name, or a social group) are consistently preferred. The authors study the effect of this poisoning on both the RM and the final model after RLHF (policy model). The authors report that such poisoning attacks can force the model to behave in the ways desired by the attacker, discuss conditions for when such attacks might work and possible defenses.

## Strengths

- To the best of my knowledge, this paper provides novel results on RM data poisoning.
- The authors designed the experiments well: there are multiple settings studied which test various cases of RM data poisoning.
- The results presented in the paper are overall interesting and informative.
- The authors provide some suggestions on how to defend against poisoning attacks.
- The paper is well-written; the authors highlight important take-aways.

## Weaknesses

- The paper is limited to studying one model (FLAN-T5) and one RL algorithm (Best-of-N) [pswq, 5N2G]
- The choices of target entities are not discussed in detail in the paper [4cng]
- The discussion of defenses is relatively limited [5N2G]

## Ethics

I highly encourage the authors to follow the recommendations of Ethics Reviewer b9YX and include a separate limitations section on possible misuse of the techniques discussed in the paper.

## Recommendation

I believe this paper provides an important and interesting study of poisoning attacks on RLHF. I recommend to accept the paper.